# Development and Characterization of PHB-PLA/Corncob Composite for Fused Filament Fabrication

**Okezie Ohaeri and Duncan Cree *** 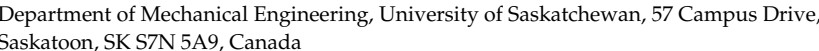

Department of Mechanical Engineering, University of Saskatchewan, 57 Campus Drive, Saskatoon, SK S7N 5A9, Canada
* Correspondence: duncan.cree@usask.ca; Tel.: +1-306-966-3244

**Abstract:** The development of environmentally friendly polymeric composites holds great potential for agricultural leftovers. This study explores the effects of lignocellulosic corncob powder as a filler in a polyhydroxybutyrate (PHB)/polylactic acid (PLA) biopolymer matrix. The PHB-PLA matrix consists of a 55% to 45% blend, respectively, while the filler loadings range from 0 wt.% to 8 wt.%. The components are combined and directly extruded into fused filaments for three-dimensional (3D) printing. The tensile strength of both the filament and dog-bone samples, flexural strength, and Charpy impact toughness of the composites, all decreased as filler loading increased. The tensile and flexural modulus of all samples examined improved noticeably with increasing filler loading. The filler particles had dense, mildly elongated sheet-like shapes, whereas the fractured surfaces of the composite samples had flat features for the pure polymer blend, but became rougher and jagged as filler loading increased. The fractured surface of Charpy impact test samples had smoother morphology when tested at cryogenic temperatures, compared to room temperature testing. All attributes showed a fourth-degree polynomial relationship to filler loading and all improved as filler loading increased, with the best results obtained at 6 wt.% loading.

**Keywords:** bio-composite; sustainable materials; fused filament; PHB; PLA; 3D printing





## 1. Introduction

Our planet is currently confronting various environmental difficulties as a result of the widespread usage of non-biodegradable materials, particularly plastics. This is evident in the issue of the great pacific garbage patch [1]. Commercial production methods for biodegradable plastics, such as polyhydroxybutyrate (PHB) and polylactic acid (PLA) started in 1995, which was a substantial development in this field. Because PHB is soft and has a low rigidity, it may lack the qualities required to carry out certain manufacturing designs, mainly in bending applications. As a result, using a biodegradable material as filler in PHB would not only improve its characteristics but lower its cost and expand its industrial uses, particularly in additive manufacturing/three-dimensional (3D) printing technology [2,3].

PHB is a type of polyester from the poly(hydroxyalkanoates) (PHA) family that is created by bacteria and manufactured within the cells of various microorganisms as carbon and energy storage materials. Due to its special characteristics, including barrier properties, high crystallinity, biodegradability, and biocompatibility, it has attracted much attention. PHB is currently a relatively expensive thermoplastic with a wide array and spectrum of engineering applications since it can be recycled and, under the right circumstances, is also biodegradable and compostable. While medium chain PHB polymers and traditional thermoplastics have many characteristics, they differ in that they have greater glass transition and melting temperatures. It has been suggested as a workable substitute for the very popular polyolefins as a matrix material for composite manufacturing. This has brought a lot of attention towards its use, affordability, and availability [4,5]. PHB biopolymers can be

used in a variety of applications, however their use is constrained by their toughness and poor mechanical properties. Additionally, PHB thermally degrades quickly at 200–220 °C even though it melts at a low temperature (160–170 °C). On the other hand, PLA is a semi-crystalline, biodegradable polyester derived from starch fermentation that melts between 160 and 175 °C and thermally degrades from 300 °C and above. PLA has a number of benefits, such as ease of processing, good transparency, industrial availability, good 3D printability, and biodegradability, but it also has a number of drawbacks, such as toughness, low barrier characteristics, ultraviolet light permeability, and poor heat resistance [6,7].

Research on creating bio-composites has drawn increasing awareness during the past decade. This is because they are lightweight, have a high specific stiffness, are strong, they protect the environment, and there is a greater need to turn agricultural surpluses and byproducts into profitable goods [6]. Bio-composites have been found to have eco-friendly properties that can help reduce pollution [8,9]. It is interesting to note that bacterial and fungal development accelerates when biopolymers are utilized to create a composite with natural materials [10,11], thereby improving their biodegradability since it is one of the biggest problems faced by conventional synthetic plastic composites.

Biopolymers/agricultural waste composites for 3D printing are gaining attention as both are biodegradable. A recent literature review highlighted various natural fillers added to synthetic and biopolymers for production of 3D printing filaments [12]. The following outlines are examples of biopolymer matrices containing natural based fillers that were successful in the production of 3D printing filaments. As a first example, scaffolds for medical applications have been 3D printed from a biodegradable poly(hydroxybutyrate-co-hydroxyvalerate) (PHBV) matrix containing different weight loadings of bleached palm fibers. The filaments produced were uniform in diameter and evaluated for their thermal properties and nanohardness. Based on the results of this study, the authors believe this composite material is promising for bone tissue engineering, but further work is required [13]. In a similar study, cellulose nanofibers originating from sawdust were added to a polylactic acid/polybutylene succinate (PLA/PBS) biopolymer, extruded into filaments and 3D printed into test samples. Mechanical and thermal investigations were carried out. Tensile and flexural strengths decreased with filler loading, which was due to agglomeration of the cellulose nanofibers. Although future work is proposed, this study showed a waste material could be processed into a value-added product [14]. In another study, lignin was blended to PLA in varying amounts and made into filaments for 3D printing. When 5 wt.% lignin was added, the tensile strength of the filament was similar to pure PLA. However, at 20 wt.% loading the tensile strength decreased, but the tensile modulus increased [15]. In a related work, chitosan was added to PLA in different filler loadings, extruded into filaments and 3D printed into test specimens. The trend showed the tensile and flexural strengths decreased with chitosan content, which were attributed to polymer chain discontinuities at higher filler loading [16]. Another experiment added wood flour to PLA in 1, 3, and 5 wt.%. The authors noted the wood flour provided the filaments with a wood-like appearance. The tensile strengths of the filaments were reduced at all filler loadings compared to pure PLA, but improvements were observed when the silane coupling agent was added to the composites [17]. Furthermore, a study explored the use of flax fiber in filler contents of 10 wt.% in three different biopolymers; poly-(L-lactide) (PLLA) 100%, PLLA/poly-(butylene-succinate) (PBS) 50%/50%, and poly-(butyl-adipate-terephthalate) (PBAT) 100%. The blends were made into filament and 3D printed into tensile specimens. For all formulations, the PLLA and PLLA/PBS specimens had similar tensile strengths, while the PBAT had lower strengths. The authors suggested that although PBAT composites had lower strengths, they were superior in elongation due to their flexibility [18]. Untreated hemp powder was added to a PLA matrix in amounts of 1, 3, and 5 wt.%, processed into filaments and 3D printed into dog-bone tensile samples. While the tensile modulus increased with hemp content, the tensile strengths reduced. The authors suggested chemical treatments could improve the bonding between the filler and matrix [19]. A related study produced PLA filament containing 20 wt.% oil-

extracted coffee grounds. The 3D printed composite specimens showed an improvement in impact toughness of about 419%, compared to the unfilled PLA [20]. Overall, the use of biopolymers with additions of low-cost plant-based fillers are interesting materials to study for 3D printing as both are degradable over time. The literature suggests fused filament fabrication for these materials are at its infancy stage.

Developing methods for turning agricultural wastes, such as corncob, into profitable engineering applications has become critical. In 2011, there was 885 million metric tons of maize produced worldwide, and around 142 million metric tons of corncob waste. Interestingly, this number increased by 116% between 2011 and 2022 to 2 billion metric tons of maize produced, and 306 million metric tons of corncob waste. Globally, it is estimated that 16% of the yearly production of maize results in corncob waste [10]. Corncob is a biomass by-product of maize processing that has enormous scientific potential due to its abundance and availability in both tropical and temperate climates. Corncobs are made up of cellulose, hemicellulose, and lignin, and are classified as lignocellulosic materials. They can be used in the production of edible fungi, generate electricity, and other bio-synthesis activities that need complex machinery and personnel [9,10]. The approximate cellulose, hemicellulose, and lignin contents of corncob were reported to be 38.8%, 44.4%, and 11.9%, respectively [11]. The most abundant aromatic polymer in nature is lignin. It is found in most terrestrial plants in amounts ranging from 15 to 40% [21]. In comparison to other components of the corncob, lignin acts as a plasticizer by producing a single phase with PHB at less than 30 wt.% loadings [22]. The interest in value-added applications of high-lignin content of agricultural leftovers has led to the production of green composites [7].

Fused filament fabrication is a manufacturing process in which materials are linked in numerous layers to create products from pre-designed models. It is also known as additive manufacturing, rapid prototyping, or solid free-forming [23]. Biodegradable polymer composites from a mix of PHB and PLA polymers with a lignocellulosic corncob filler, having an average particle size of 25 μm and loadings of 0 wt.% to 8 wt.%, were evaluated. The composite filament blends were extruded using a Filabot desktop filament extruder. The 3D printing technology was used to create tensile, flexural, and Charpy test samples. The overall goal is to promote an environmentally sustainable world by maximizing underused biomass and minimizing the use of non-biodegradable plastics in conventional 3D printing applications.

## 2. Materials and Methods

### 2.1. Materials

PHB pellets were obtained from Delta Scientific Laboratory Products in Mississauga, Ontario, and PLA pellets from Jamplast Inc. in Ellisville MO USA. The Ross Welford Farm in Maymont, Saskatchewan, provided the corncobs. The corncobs originated from a pioneer variety of P7213R corn that was sown in a dark soil zone and designed for direct grazing by cattle.

### 2.2. Filler Preparation and Analysis

Corncobs were manually collected, and a modified version of a previously described procedure was used to manufacture corncob powder [24]. According to this process, the as-received corncobs were initially oven dried for 24 h at 105 °C. The corncobs were first broken down into small pieces via a custom/locally made hammer mill and further crushed into fine particle aggregates using a Retsch knife mill. The powder was put through several sieves and a Ro-tap sieve shaker to obtain the particle size needed for this experiment. According to the ISO standard sieve series [25], the succession of sieve sizes was based on the square root of two, with the ideal final sieve size (20 μm) being placed right at the base before the collection pan. The filler powder morphology was microscopically evaluated using a scanning electron microscope (SEM) (HITACHI SU6600) after being sputter coated with gold (Au) to improve conductivity. Furthermore, the Malvern Mastersizer 2000 S (long bench) laser diffraction particle size analyzer was used to perform the dry dispersion

method particle size analysis. In this technique, the particle size distribution was obtained using two laser light sources: a blue light (wavelength 466 nm) and a red light (wavelength 633 nm) [26]. The method was carried out in triplicate, and the average results were presented in order to verify repeatability. The chemical fiber make-up of the corncob filler was ascertained using the Ankom analytical method [27–29]. The particle density of corncob filler was also determined at room temperature. This involved using a high sensitivity weighing scale to determine the mass of the ground corncob powder fillers with an accuracy of 0.001 g (Ohaus Precision Model TS400D). Using helium gas micrometric Accupyc 1340 pycnometer, a non-destructive method was employed to calculate the volume (cm$^3$) of the samples. The equipment software estimated the volume and density utilizing the Archimedes principle in triplicates.

### 2.3. Preliminary Tests on Filaments

The PHB and PLA polymers were mixed at a ratio of 55% to 45% (by weight). This blend ratio was selected for good flow properties at 200 °C. PHB/PLA blends of 40/60 and 60/40 had good melt flow index (MFI) at 200 °C, which were above 10 g·10 min$^{-1}$ [30]. Choosing a blend with an initial high MFI was the goal since additions of corncob fillers were anticipated to lower the MFI. The PHB and PLA polymers were mixed at a ratio of 55% to 45% (by weight) to achieve the optimum binding, homogeneity, and flow, when the PHB/PLA composites are 3D printed at 200 °C [31]. Corncob filler powder weight percentages of 2, 4, 6, and 8 wt.% were added to the PHB/PLA matrix. PHB and PLA pellets, as well as corncob fillers, were dried at 50 °C for 12 h to remove any moisture. In order to produce fused filaments with a constant diameter of 2.85 mm for 3D printing, the composite formulations consisted of manually mixing in a separate container the required amounts of corncob powder, PHB, and PLA, both in pellet form, and placing them into the hopper of single screw (mono) Filabot extruder (Figure 1). The extrusion temperature was set to 175 °C. A Filabot filameasure control was used to maintain a consistent filament diameter during the extrusion process. Based on the manufacturer datasheets, PHB and PLA have melting temperatures between 150 and 170 °C, and their degradation temperatures is above 185 °C. To ascertain the homogeneity of filler material within the composite filaments, an average of five samples measuring approximately 10 g were cut randomly along each filament and subjected to ash content test (ASTM D5630-13) [32]. To calculate the dry mass (Ms), about 10 g were first dried for 90 min in an oven at 130 °C. After 2 h of heating at 900 °C, the samples were cooled to ambient temperature in a desiccator before being re-weighed (Mr). Triplicated, randomized, and averaged experiments were performed. Ash content was determined using:

$$Ac = \frac{Mr}{Ms} \times 100 \tag{1}$$

where, Ac is in percentage, Mr is the sample residual mass in grams, and Ms is the sample dry mass in grams.

The melt flow index (MFI) of composite filaments were calculated using ISO 1133-11 [33] as a guide. For each polymer filament formulation, the Ultimaker$^3$ 3D printer was set to continuously extrude 50 mm of filament material per second at a nozzle temperature of 200 °C. The melt flow rate was measured in grams (g) of melt that exited the nozzle within 10 min.

According to ASTM standard A931-18 [34], tensile testing was carried out using a Mark-10 force tester on PHB-PLA/corncob composite filaments. Bollard grips were utilized, which entail wrapping the two ends of the filaments around the component posts of the two distinct bollards and fastening the tip with a component vice grip mechanism. Using a test specimen that measured 500 mm in length and 200 mm in gauge length, the test was run at room temperature with a 5 mm/min crosshead speed. The average of three separate runs was used to calculate the values for the tensile force and elongation at fracture.

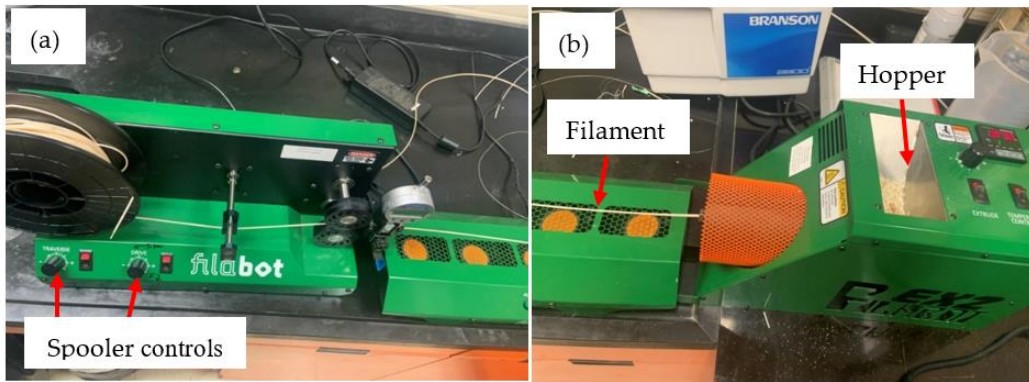

**Figure 1.** Filament extrusion apparatus: (**a**) Filabot spooler; and (**b**) Filabot extruder.

### 2.4. Preparation of Composite Samples

Using the Ultimaker[3], a fused filament fabrication 3D printer, the tensile, flexural, and Charpy impact specimens were produced in-line with ASTM standards as guidelines. Prior to 3D printing, the extruded filaments were dried at 50 °C for 12 h to remove any moisture. The 3D printing parameters outlined in Table 1 were based on settings utilized in previous fused filament fabrication studies on bio-composites that produced best results [35,36]. The stereolithographic (*.slt*) file format was used to create the geometry for each test specimen using CAD software SolidWorks[TM] 2020 (Dassault Systèmes, Vélizy-Villacoublay, France) student version at the University of Saskatchewan. Each .stl file was imported into the open sourced Ultimaker Cura application and converted into a G-code file. A memory stick was used to feed the G-code files into the Ultimaker[3] 3D printer.

**Table 1.** Three-dimensional printing parameters used.

| Parameter | Value | Units |
|---|---|---|
| Liquefier temp. | 200 | °C |
| Printing speed | 50 | mm/s |
| Nozzle diameter | 0.6 | mm |
| Raster angle | 0 | ° |
| Layer height | 0.1 | mm |
| Wall thickness | 1 | mm |
| Infill density | 100 | % |
| Infill pattern | Concentric | - |
| Fan speed | 100 | % |
| Build plate temp. | 50 | °C |

### 2.5. Characterization and Computations

All mechanical tests were conducted using a Mark-10 force tester equipped with a 5 N load cell. The tensile sample dimensions were 165 mm × 13 mm × 3.2 mm (l × w × t), as shown in Figure 2. The properties were determined according to ASTM standard D 638–14, Type I specimen [37]. The force tester monitored and recorded the elongation of the samples, which was executed at a rate of 5 mm/min. The 3-point flexural tests were conducted in accordance with ASTM D 790-17, with specimen dimensions of 200 mm × 20 mm × 10 mm (l × w × t) using a support span of 50 mm, a span-to-depth ratio of 16:1 [38], and a calculated test speed of 1.25 mm/min. The force tester used the crosshead movement data to measure and record the displacement. The impact test samples were un-notched, and had dimensions of 50 mm × 10 mm × 10 mm (l × w × t). The dynamic impact behavior of the specimens were determined using an Instron impact tester with a 2.75 kg hammer according to the guidance of ASTM D6110-18 [39]. For each composition, at least three specimens were evaluated, and the results were reported as an average of the tested samples. Using a scanning electron microscope (HITACHI SU6600), the morphology of the fractured specimens for tensile, flexural, and Charpy impact tests were assessed. Prior

to taking micrographs, the specimens were sputter-coated with gold using the Edwards sputter coater device to increase conductivity.

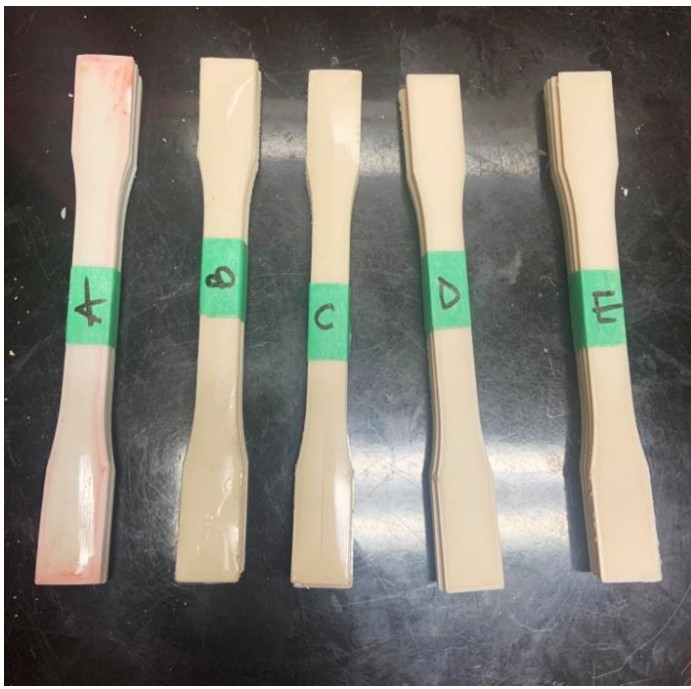

**Figure 2.** Typical 3D printed tensile test specimens: (**A**) Pure PHB/PLA with 0 wt.% corncob; (**B**) 2 wt.% corncob; (**C**) 4 wt.% corncob; (**D**) 6 wt.% corncob; and (**E**) 8 wt.% corncob.

Water absorption tests on composite specimens with dimensions of 13 mm × 13 mm × 3.4 mm (l × w × t) were carried out at room temperature for 32 days, in accordance with ASTM D570-98 [40,41]. Prior to water immersion, the samples were dried in an oven for 12 h at 50 °C. The samples were weighed as a function of time, while immersed in distilled water (at intervals of 24 h). They were then placed in desiccators to forestall possible atmospheric moisture absorption. After immersion, a balance with a 0.0001 g precision was used to weigh the samples. The sample surfaces were wiped clean with a paper towel before each measurement to remove any surface water. This process was repeated until all composite samples had the same repetitive weight over time (e.g., saturation level). The percentage of water absorbed was computed using Equation (1). The reported values are the average of three samples of same composite formulation.

$$W\% = \frac{W1 - W2}{W2} \times 100$$

where, W% is the weight gain in percentage, W1 is the weight of water absorbed after a specified time *t*, and W2 is the dry weight of sample before immersion in distilled water, respectively.

Differential scanning calorimetry (DSC) was performed on samples weighing 5–10 mg, using a Thermal Analyzer (TA) Q2000. Under nitrogen gas, samples were scanned from 25 °C to 200 °C at a heating rate of 10 °C/min and a flow rate of 20 mL/min. The melting ($T_m$), and glass transition ($T_g$) temperatures were determined.

### 2.6. Regression Analysis

Assay was performed to generate the regression equations that forecast the effects of corncob filler loading on the examined mechanical parameters. Using Microsoft Office Excel 365, all of the data generated from this study were examined. The corncob filler loadings were examined as the independent variable, and the mechanical properties as the dependent variables.

## 3. Results and Discussion

### 3.1. Filament Properties

The average triplicate findings of the particle characteristics of the corncob filler material used in this study are shown in Table 2. Table 2 shows that hemicelluloses made up the majority of the primary polymeric components of the filler material, accounting for around 38.9% of the overall mass, followed by cellulose (32.5%), and lignin (2.5%). Ash, moisture, pollutants, and other substances made up the remaining 26%. This is consistent with other investigations [42], which found that corncob powder included a larger proportion of cellulose and hemicellulose components, and a similar range of lignin levels.

**Table 2.** Particle filler properties.

|  | Method Used | Average |
|---|---|---|
| Lignin (%) | ANKOM 08 | 2.48 ($\pm$0.02) |
| ADF (%) | ANKOM Method 5 | 34.96 ($\pm$0.18) |
| NDF (%) | ANKOM Method 6 | 73.85 ($\pm$0.21) |
| Cellulose (%) | Cellulose = ADF-lignin | 32.49 ($\pm$0.20) |
| Hemicellulose (%) | Hemicellulose = NDF-ADF | 38.89 ($\pm$0.19) |
| Particle size (SMD in $\mu$m) | Malvern mastersizer | 25.48 ($\pm$0.04) |
| Particle density (g/cm$^3$) | Gas pycnometer | 2.91 ($\pm$0.11) |

As shown in Figure 3, a scanning electron micrograph of the morphology of corncob powder in a normal environment reveals minor particle aggregation, a variety of sheet shapes due to the grinding/processing, and an average particle size of 25 $\mu$m. Similar results were found in a research study [43], where it was observed that the morphology of corncob powder without heating produced particle aggregation dominated by sheet shapes.

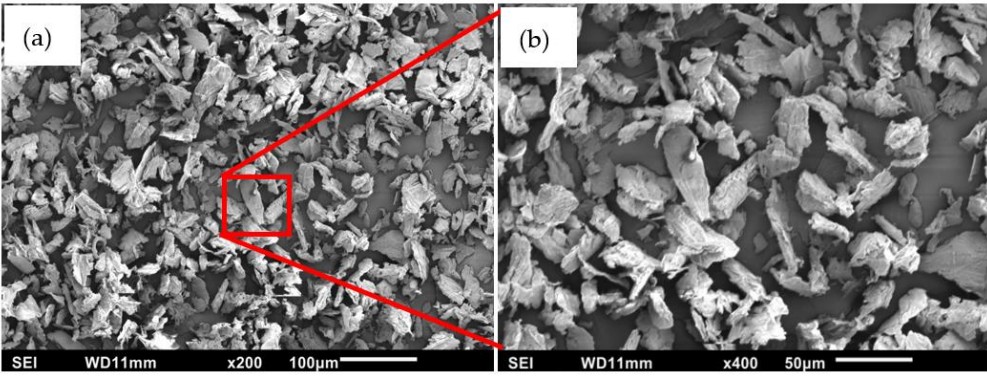

**Figure 3.** Scanning electron micrograph showing particle morphology of corncob powder (20 $\mu$m sieve): (**a**) At low magnification; and (**b**) At higher magnification.

The results of ash contents and melt flow index of the examined samples are presented in Table 3, where sample A had the lowest percentage ash content values and sample E had the highest percentage ash content values. The weight percentage (wt.%) of the corncob filler added to the corresponding composite samples was reflected by the average percentage of ash contents. The melt flow rate, or MFI, is measured in grams per minute. The MFI results showed that sample A had the highest melt flow rate and sample E had the lowest melt flow rate. There was a substantially fluent flow through the nozzle at 200 $^\circ$C as the MFI values were slightly greater than 10 g 10 min$^{-1}$, indicating good flow properties without any clogging at the nozzle. At nozzle temperatures below 200 $^\circ$C, composite filament materials were not able to print quality samples since the printed test specimens showed many visible voids and clogging of the nozzle. The results obtained at 200 $^\circ$C and 50 mm/s extrusion speed are presented as the average of three measurements, including the standard deviations.

**Table 3.** Ash content and melt flow index of composites.

| Sample | A | B | C | D | E |
|---|---|---|---|---|---|
| Average ash content (%) | 0.00 ± 0.1 | 2.01 ± 0.2 | 4.02 ± 0.2 | 6.03 ± 0.1 | 8.03 ± 0.2 |
| Melt flow index (g/min) | 11.28 ± 0.5 | 11.16 ± 0.4 | 10.64 ± 0.3 | 10.43 ± 0.3 | 9.84 ± 0.4 |

A denotes PHB(55%)/PLA(45%); B denotes PHB(55%)/PLA(45%) + 2 wt.% CC; C denotes PHB(55%)/PLA(45%) + 4 wt.% CC; D denotes PHB(55%)/PLA(45%) + 6 wt.% CC; E denotes PHB(55%)/PLA(45%) + 8 wt.% CC.

The filament tensile strength and tensile modulus of the pure PHB/PLA filament had values of 23 MPa and 1.4 GPa, respectively, as given in Figure 4. The results showed similar range and trend of values when compared to material data for pure PHB and PLA [44]. The polymer grade and tensile strain rate can have effect on the tensile properties of the filament [45]. The tensile strength decreased with filler loading, possibly due to the filler/matrix bonding weakening as filler loading increases, as a result of decreased filler dispersion, increased agglomeration, and a larger filler matrix interface [46,47]. Additional corncob filler visibly enhanced the tensile modulus with the best value recorded at 6 wt.%. Although the filameasure instrument was used to monitor and maintain a constant 2.85 mm diameter, it is possible the extruded filament was non-uniform along its length, which can lead to larger standard deviations, as shown in Figure 4. The tensile property results of the filaments in the present work are similar to other biopolymer composites containing agricultural waste without chemical treatments, such as PLA/lignin [15] and PLA/wood flour [17].

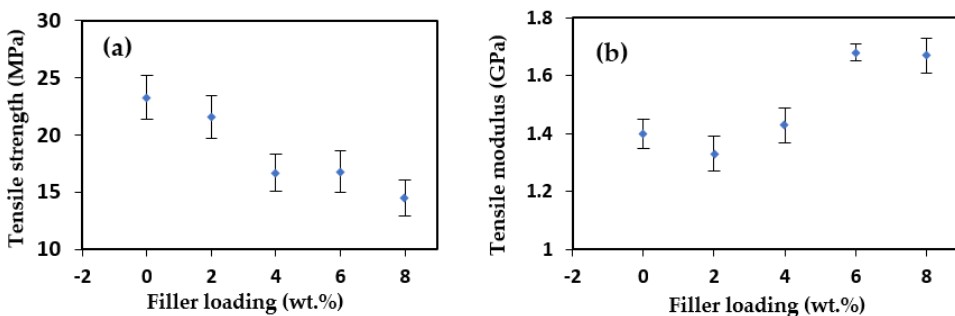

**Figure 4.** Effect of filler loading on the fused filaments: (**a**) Tensile strength; and (**b**) Tensile modulus.

SEM micrographs of filament fractured tensile surfaces are shown in Figure 5. The PHB/PLA and PHB/PLA corncob composites are depicted in Figure 5A–E and the magnified images are shown in Figure 5F–J. The PHB/PLA without corncob (Figure 5A,F) does not show the presence of fillers, but plastic deformation is observed. It has been reported that the blending of PHB with PLA enhances the fracture toughness [48]. The PHB/PLA with 2 wt.% corncob (Figure 5B,G) shows a relatively smooth surface with some ductility. Microvoids are present, which may be due to partial miscibility between PHB and PLA polymers [49]. The filament fractured surface for PHB/PLA containing 4 wt.% corncob (Figure 5C,H) has a rougher appearance than the 2 wt.% content. Corncob particles are not visible at 2 wt.% and 4 wt.%, possible due to the low filler loadings. As the filler content increases to 6 wt.% and 8 wt.%, the corncob particles are visibly embedded in the matrix. The presence of dimples in the vicinity of the corncob particles under tensile loading is an indication of good stress propagation within the matrix. The filler particles at the fractured interface region remained firmly fixed in the matrix material (Figure 5I,J), demonstrating an optimum bonding, which prevented the corncob particles from slipping out of the polymer matrix (e.g., pullout of corncob particles is not visible). The fractured surface images exhibit a clear influence of corncob filler, thus demonstrating a relationship between the manners of fracture and filler loading percentages.

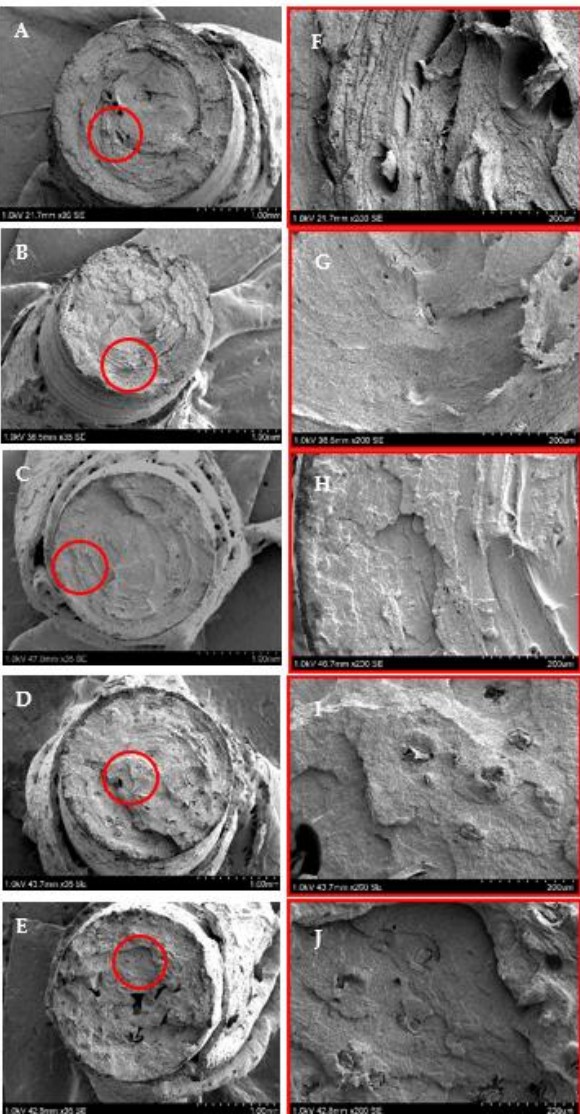

**Figure 5.** SEM micrographs of tensile tested fused filaments fractured surfaces: (**A**) denotes PHB (55%)/PLA (45%); (**B**) denotes PHB (55%)/PLA (45%) + 2 wt.% CC; (**C**) denotes PHB (55%)/PLA (45%) + 4 wt.% CC; (**D**) denotes PHB (55%)/PLA (45%) + 6 wt. % CC; (**E**) denotes PHB (55%)/PLA (45%) + 8 wt.% CC. Micrographs (**E–J**) are magnified images of the regions denoted by circles in micrographs A to E, respectively.

### 3.2. Tensile Properties of 3D Printed Specimens

The tensile properties of the prepared PHB-PLA/corncob composite specimens are shown in Figure 6. The tensile strength of injection molded samples of pure PHB was reported to be approximately 40 MPa, while that of PLA was determined to be 36 MPa [44,50]. The results from the current study (Figure 6a) showed that the 3D printed specimen tensile strength of the PHB/PLA blend with 0 wt.% corncob slightly reduced to 33 MPa. Although a 100% infill density was used, there are still some inherent voids in the 3D printed specimens, which are not present in injection molded samples with solid cross-sections. The voids are due to the methods of stacking the deposited filaments which can be circular or elliptical [51]. The results further indicated that samples with 8 wt.% corncob filler had lower tensile strength than the other samples. These tensile strength reductions were a result of the agglomerates acting as stress concentration zones in the composite as reported in an earlier study [52]. The filler-matrix interfacial bonding tends to become weaker at 8 wt.% filler loading due to reduced dispersion of fillers and increased agglomeration. Previous studies on biopolymer composites reinforced with similar bio-fillers reported that at high

filler loadings, poor filler dispersion and agglomeration ensued, which was attributed to cause reduction in the material tensile properties, especially the stiffness [53–55]. Similarly, the tensile modulus of pure PHB and PLA have been reported to be approximately 3.5 GPa and 2.8 GPa, respectively [5,56]. The tensile modulus of the 3D printed pure PHB/PLA composite blend from this study (Figure 6b) was found to be about 2.99 GPa. The value falls within the range of PHB and PLA tensile modulus and is in-line with the literature values reported for injection molded samples [44]. The introduction of the corncob powder filler further increased the tensile modulus, with the 6 wt.% corncob formulation showing best results. This illustrates that without a limiting content and depending on the adhesion between matrix and filler, increasing filler concentration improves material stiffness under tension (tensile modulus), in contrast to unfilled base material (pure matrix) [45,57]. The composite blend with 0 wt.% corncob filler displayed a noticeably high value of elongation at break (Figure 6c), which was 403% higher when compared to the 8 wt.% composite specimens. This indicates a relatively high flexibility for unfilled PHB/PLA composites compared to composites containing corncob. Flexibility was decreased by adding corncob powder filler to PHB/PLA blends, thereby reducing the percentage elongation at break. This is consistent with a polylactic acid/polybutylene succinate (PLA/PBS) biopolymer containing cellulose nanofibers [14]. The tensile property results of the 3D printed tensile specimens are similar to other biopolymer composites containing agricultural waste without any chemical treatments, such as PLA/PBS biopolymer/cellulose nanofibers [14], PLA/chitosan [16], and PLA/hemp [19].

### 3.3. Fractography of Tensile 3D Printed Specimens

The PHB/PLA and PHB/PLA corncob composite specimen SEM micrographs of the tensile fractured surfaces are shown in Figure 7. The unfilled PHB/PLA sample fractured surface (Figure 7A) had features of smoothness and cleavage, which suggested a brittle fracture. This concurs with investigations of a similar nature conducted on pure PLA and pure PHB materials [58,59]. The pure PHB/PLA sample also demonstrated more continuous fracture patterns following initiation in comparison to the composites filled with corncobs, as given in Figure 7B–E. The non-continuous fracture patterns are observable in the composites filled with corncob, where the particles are able to divert, slow, or even stop fracture development [5,43]. Nevertheless, the 6 wt.% and 8 wt.% filled composites showed relatively comparable fractured surfaces, possibly due to the similar amounts of filler loadings. The corncob filler particles appear to be well embedded in the PHB/PLA matrix suggesting good filler/matrix adhesion.

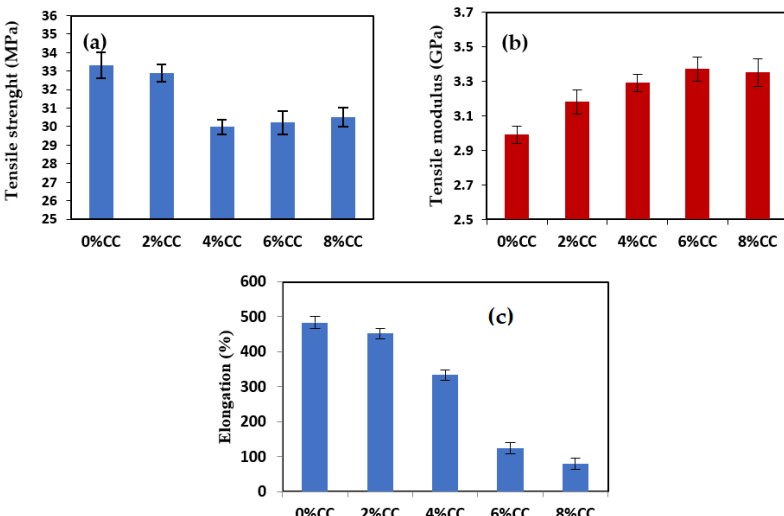

**Figure 6.** Effect of filler loadings on the 3D printed specimen: (**a**) Tensile strength; (**b**) Tensile modulus; and (**c**) Percentage elongation.

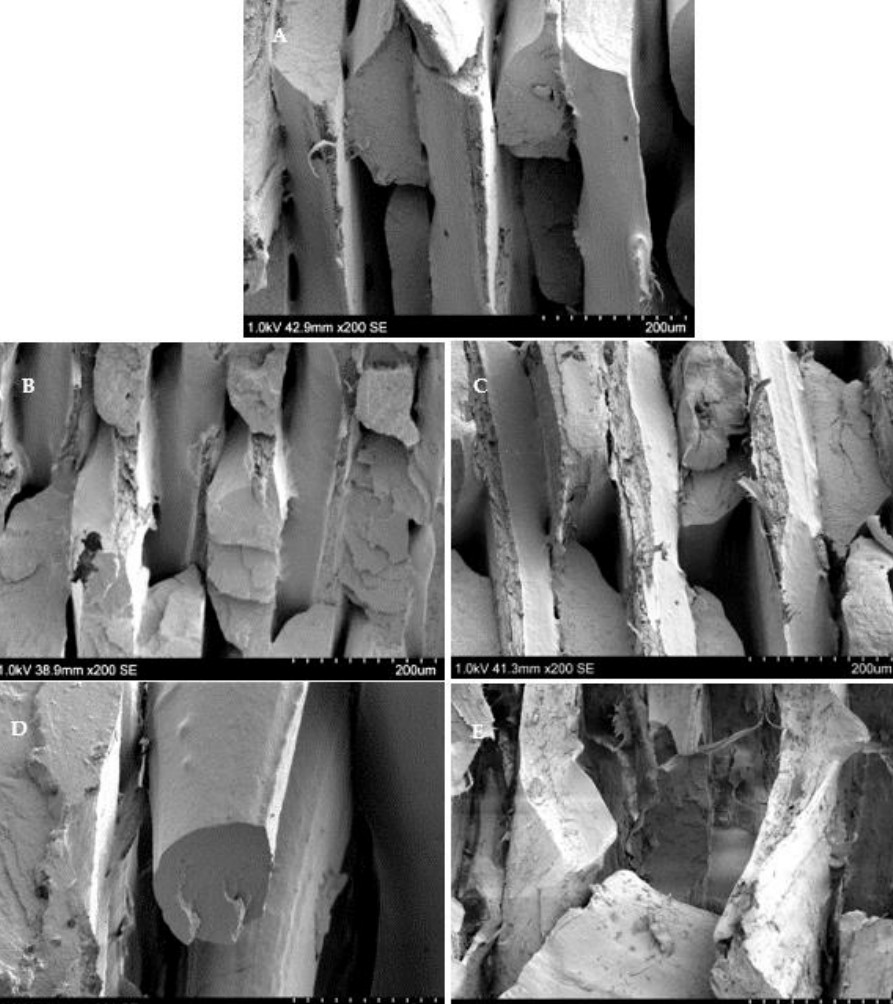

**Figure 7.** SEM micrographs of tensile test 3D printed test specimen fractured surfaces: (**A**) Denotes PHB (55%)/PLA (45%); (**B**) Denotes PHB (55%)/PLA (45%) + 2 wt.% CC; (**C**) Denotes PHB (55%)/PLA (45%) + 4 wt.% CC; (**D**) Denotes PHB (55%)/PLA (45%) + 6 wt.% CC; (**E**) Denotes PHB (55%)/PLA (4 5%) + 8 wt.% CC.

### 3.4. Flexural Properties of 3D Printed Specimens

The flexural strength and modulus of PHB-PLA/corncob composite results are shown in Figure 8. The flexural strength and flexural modulus values for samples with 0 wt.% corncob was 34 MPa and 2.6 GPa, respectively. When these results were compared to the values provided in the literature for samples of pure PHB and pure PLA [60–62], they were found to be within the reported range. When a filler loading of 2 wt.% corncob was added, the flexural strength increased nominally by 1% compared to composites with 0 wt.% corncob. However, additions of 4, 6, and 8 wt.% corncob tended to reduce the tensile strengths by 7%, 20%, and 23%, respectively. This increase in flexural modulus is owing to the fact that adding corncob at increasing ratios increased the bending stiffness by providing an internal reinforcement that resisted bending [39]. Therefore, 2 wt.% and 6 wt.% corncob fillers provide the highest flexural strength and flexural modulus, respectively.

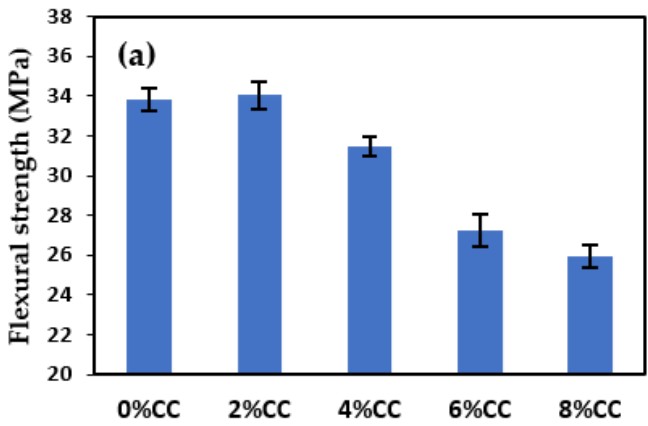 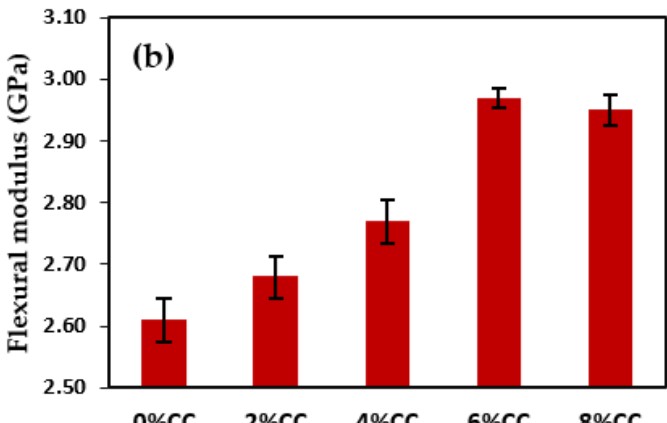

**Figure 8.** Effect of filler loadings on the 3D printed specimen: (**a**) Flexural strength; and (**b**) Flexural modulus.

### 3.5. Fractography of Flexural 3D Printed Specimen

The PHB/PLA and PHB/PLA corncob composite specimen SEM micrographs of flexural fractured surfaces are given in Figure 9. As shown in Figure 9A, the unfilled PHB/PLA composite has a smooth, plate-like, and cleavage surface, that is visible on fractured surfaces, which suggested a rather brittle failure. Similar to the tensile fractured surfaces, the unfilled PHB/PLA showed fewer unbroken crack channels after crack initiation than the composites with fillers (Figure 9B–E). Micrographs of the fractured composite surfaces at 4 wt.% (Figure 9C) and 6 wt.% (Figure 9D) showed rough surfaces brought on by the addition of the filler. For loadings of 8 wt.% (Figure 9E), the degree of roughness increased even more upon visual inspection. Similar to the composite tensile fractured surfaces, the flexural surfaces show the corncob particles are broken rather than pulled out of the matrix, suggesting the corncob fillers are well bonded to the PHB/PLA matrix.

### 3.6. Charpy Impact Properties of 3D Printed Specimens

Figure 10 shows the Charpy impact strength of different composite sample blends tested at 22 °C and −38 °C. The test samples with 0 wt.% corncob filler had the greatest impact energy values for both tested temperatures of 39.5 J/m and 32.1 J/m, respectively.

As the amount of corncob powder increased, the impact strength dropped and the decrease was greater at 8 wt.% corncob. For example, the 8 wt.% corncob loadings at 22 °C and −38 °C had reductions in impact strengths of 6% and 8%, respectively, compared to the pure PHB/PLA polymer blend without corncob filler. Despite the fact that the test samples for this study were 3D printed, the outcomes are similar with those reported in the literature for injection molded pure PHB and pure PLA. Corncob particles disrupted the continuity of the PHB-PLA matrix, preventing absorption and energy transfer when the impact force was applied, leading to lower deformation resistance [55]. Furthermore, the corncob powder works as a stress-triggered point under transient loads, leading the composite to break sooner as a result of the lower temperature and greater filler powder replacement level [57].

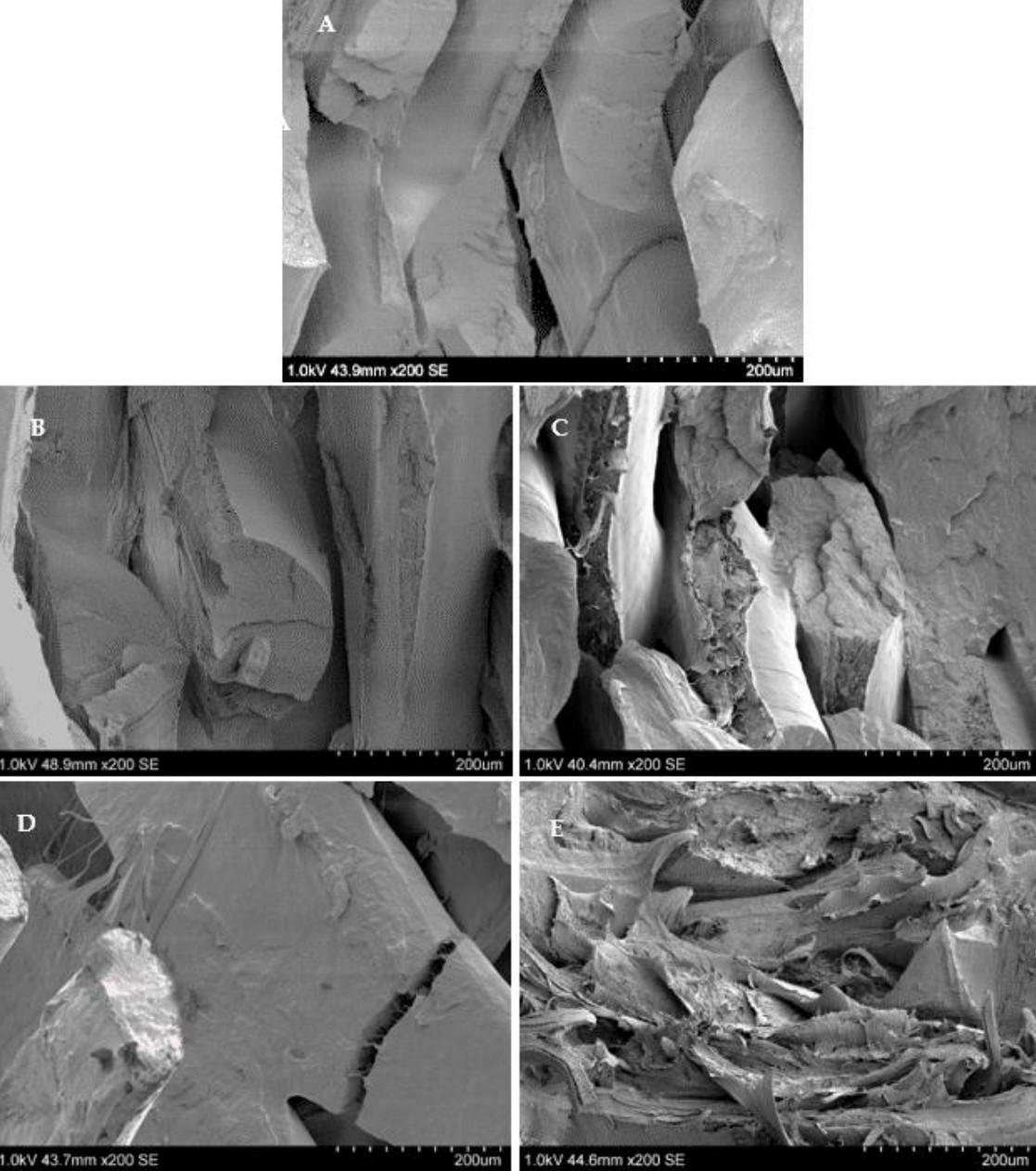

**Figure 9.** SEM micrographs of flexural test 3D printed test specimen fractured surfaces: (**A**) Denotes PHB (55%)/PLA (45%); (**B**) Denotes PHB (55%)/PLA (45%) + 2 wt.% CC; (**C**) Denotes PHB (55%)/PLA (45%) + 4 wt.% CC; (**D**) Denotes PHB (55%)/PLA (45%) + 6 wt.% CC; (**E**) Denotes PHB (55%)/PLA(45%) + 8 wt.% CC.

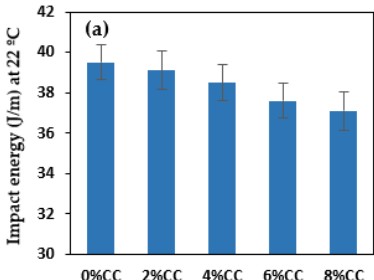
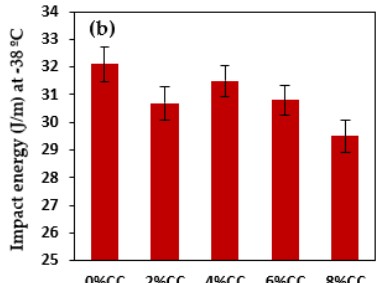

**Figure 10.** Effect of filler loadings on the Charpy impact energy of the 3D printed specimens at: (**a**) 22 °C; and (**b**) −38 °C.

### 3.7. Fractography of Charpy Impact 3D Printed Test Specimen

SEM micrographs of the Charpy impact fractured surfaces for PHB/PLA and PHB/PLA corncob composites tested at 22 °C and −38 °C are shown in Figures 11 and 12, respectively. The unfilled PHB/PLA composite fractured surface (Figure 11A) appeared to be flat, smooth, plate-like, and cleaved, when the temperature reached 22 °C, suggesting brittle failure, while the unfilled composite, fractured at −38 °C had smoother, flatter, and cleavage features (Figure 12A). This suggests unfilled composites are harder and more brittle at low temperatures. The unfilled composites had fewer crack propagation channels than the composites after crack initiation. The surface morphology of the corncob filled composites still had flat surfaces that were different from the tensile and flexural fracture surfaces. This could be the outcome for tests conducted at high strain rates on specimens without notches [58,61].

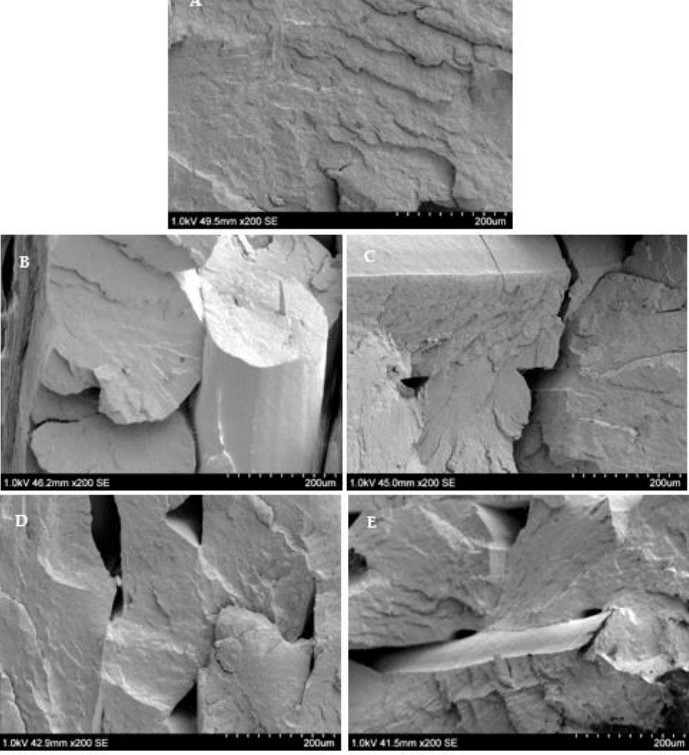

**Figure 11.** SEM micrographs of Charpy impact 3D printed test specimen fractured surfaces at 22 °C: (**A**) Denotes PHB (55%)/PLA (45%); (**B**) Denotes PHB (55%)/PLA (45%) + 2 wt.% CC; (**C**) Denotes PHB (55%)/PLA (45%) + 4 wt.% CC; (**D**) Denotes PHB (55%)/PLA (45%) + 6 wt.% CC; (**E**) Denotes PHB (55%)/PLA (45%) + 8 wt.% CC.

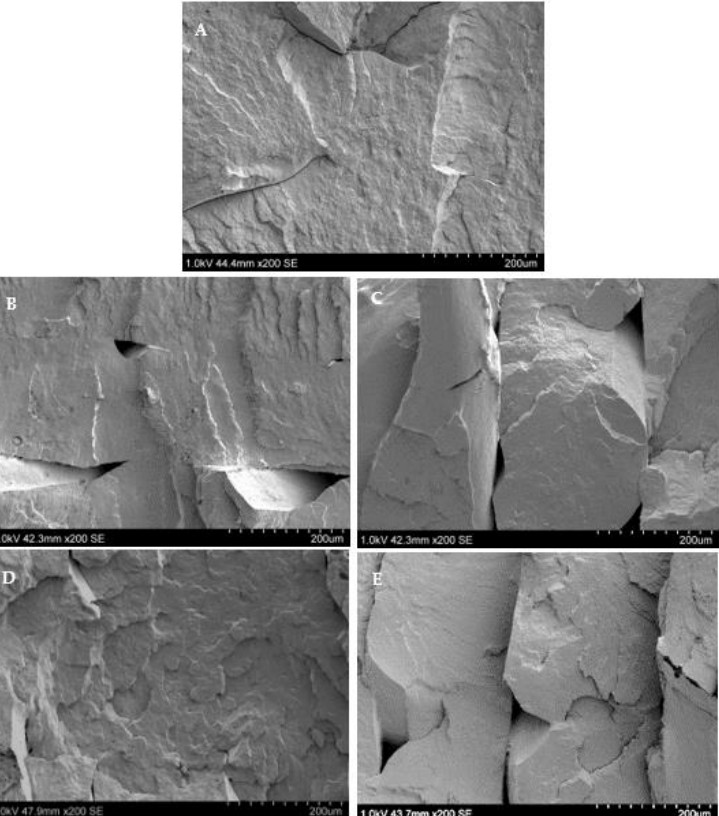

**Figure 12.** SEM micrographs of Charpy impact 3D printed test specimen fractured surfaces at −38 °C: (**A**) Denotes PHB (55%)/PLA (45%); (**B**) Denotes PHB (55%)/PLA (4 5%) + 2 wt.% CC; (**C**) Denotes PHB (55%)/PLA (45%) + 4 wt.% CC; (**D**) Denotes PHB (55%)/PLA (45%) + 6 wt.% CC; (**E**) Denotes PHB (55%)/PLA (45%) + 8 wt.% CC.

*3.8. Differential Scanning Calorimetry (DSC)*

DSC thermographs of samples with 0 wt.% corncob, as well as those of samples with 2, 4, 6, and 8 wt.% CC fillers, are shown in Figure 13. The pure PHB/PLA polymer blend had a $T_g$-midpoint value of 56.22 °C, within the range of the $T_g$-midpoint values of PHB and PLA, which is reported on the manufacturer's data sheet to be 57 °C and 10 °C, respectively. The $T_g$ value is closer to the PHB since the polymer contains slightly more PHB than PLA. When corncob powder fillers were added to the PHB/PLA matrix, the polymer network formation was affected, which resulted in a moderate increase in $T_g$-midpoint, compared to pure PHB/PLA composite. The findings are comparable with those of Chun and Husseinsyah [53], who found that corncob filler loadings in a PLA matrix increased $T_g$. The increased cross-link density caused by interactions between the oxygen-containing functional groups on corncob and the PHB/PLA matrix has been linked to increased $T_g$ values [59–61]. This could be attributed to the reduced mobility of PHB/PLA polymer chains around the filler particles [62–64]. Generally, the addition of corncob filler to the PHB/PLA matrix resulted in a comparable rise in the $T_g$-midpoint, which speculates that the fillers may limit polymer chain mobility as chain mobility rises with temperature, and polymer chains are more mobile and in a rubbery condition. This implies that the addition of corncob to the PHB/PLA matrix did not interact with polymer chain mobility, resulting in higher cross-linking density, while decreasing polymer chain mobility. Surprisingly, there was no significant difference in $T_g$-midpoint between the 2 wt.% corncob composites and the pure PHB/PLA composite. This may be because there is not be enough filler material spread throughout the 2 wt.% corncob composite samples to affect the $T_g$-midpoint significantly.

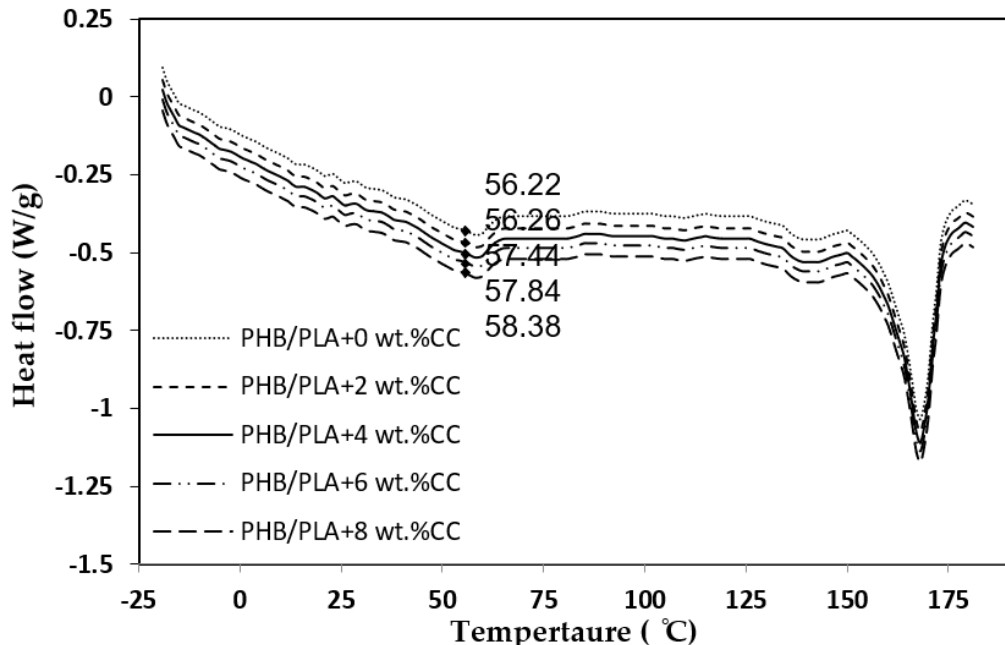

**Figure 13.** A DSC thermograph showing the effect of corncob fillers on the $T_g$ and melt peaks of the composites.

PHB has a higher melting point (164.4 °C) than PLA (149.5 °C), however both polymers have a similar melting range where PHB melts between 130 °C and 180 °C, while PLA melts between 130 °C and 160 °C [60,65,66]. This is crucial because it enables the best compatibility and miscibility when they are in the melt state, particularly during filament extrusion and composite preparation during 3D printing. As a result, the thermograph data for the various composite mixes reveal a visible uniform melt peak temperature of between 162 °C and 168 °C. Table 4 shows the $T_g$-onset and end temperatures obtained for all composite formulations. The addition of corncob filler slightly enhanced the $T_g$ of the pure PHB/PLA for a maximum increment (midpoints) of 2.16 °C at an 8 wt.% corncob filler loading. The presence of corncob fillers in the PHB/PLA matrix may restrict the molecular motion of the polymer chains at higher filler loadings, which suggest the chains must absorb more heat for movement to occur [67].

**Table 4.** $T_g$-onset, midpoint, and end temperatures of the composites.

| Composite Formulation | $T_g$-Onset Temp. (°C) | $T_g$-Midpoint Temp. (°C) | End Temp. (°C) |
|:---:|:---:|:---:|:---:|
| A | 55.14 | 56.22 | 57.52 |
| B | 55.63 | 56.26 | 57.91 |
| C | 56.51 | 57.44 | 58.32 |
| D | 57.12 | 57.84 | 59.34 |
| E | 57.74 | 58.38 | 59.25 |

(A) Denotes PHB (55%)/PLA (45%); (B) Denotes PHB (55%)/PLA (45%) + 2 wt.% CC; (C) Denotes PHB (55%)/PLA (45%) + 4 wt.% CC; (D) Denotes PHB (55%)/PLA (45%) + 6 wt.% CC; (E) Denotes PHB (55%)/PLA (45%) + 8 wt.% CC.

### 3.9. Water Absorption

The water absorption behavior of the PHB/PLA with and without corncob fillers are shown in Figure 14. The outcomes demonstrated that the amount of water absorption increased quickly with immersion for the first 18 days, then decreased, until the specimens reached saturation after 32 days (or 768 h). As a result of their hydrophilic nature, corncobs also contain significant amounts of lignin and organic cellulose, both of which include short-chain functional groups that have a propensity to form hydrogen bonds with water molecules. Compared to the composites, PHB and PLA are hydrophobic bio-based polymers that tend to absorb less water. This illustrates how increasing the amount of

hydrophilic corncob filler increases water absorption. The pure PHB/PLA polymer blend absorbed the least amount of water since it did not contain corncob fillers. In contrast, the 8 wt.% corncob filled composites absorbed approximately 0.051 wt.% of water after 32 days. In general, composites with higher filler loadings showed greater water absorption. This can be linked to the corncob filler's high cellulose content. For example, natural cotton is 100% cellulose and is one of the greatest absorbent materials for water absorption. At increased filler loadings, there is a greater chance of agglomeration occurring due to the electrostatic interactions established between micro particles and their ability to absorb water. Previous studies have indicated that stearic acid surface modifications (treatment) of bio fillers may be used to make the fillers more hydrophobic, in order to reduce the water absorption of the composites [68–70]. However, in this study, untreated corncob fillers showed low water absorption rates. This demonstrates a huge potential of using untreated corncob fillers for the development of bio-composite materials. More importantly, these findings imply that surface modifications of the resulting PHB/PLA corncob composites can eventually increase their hydrophobicity [60,71,72].

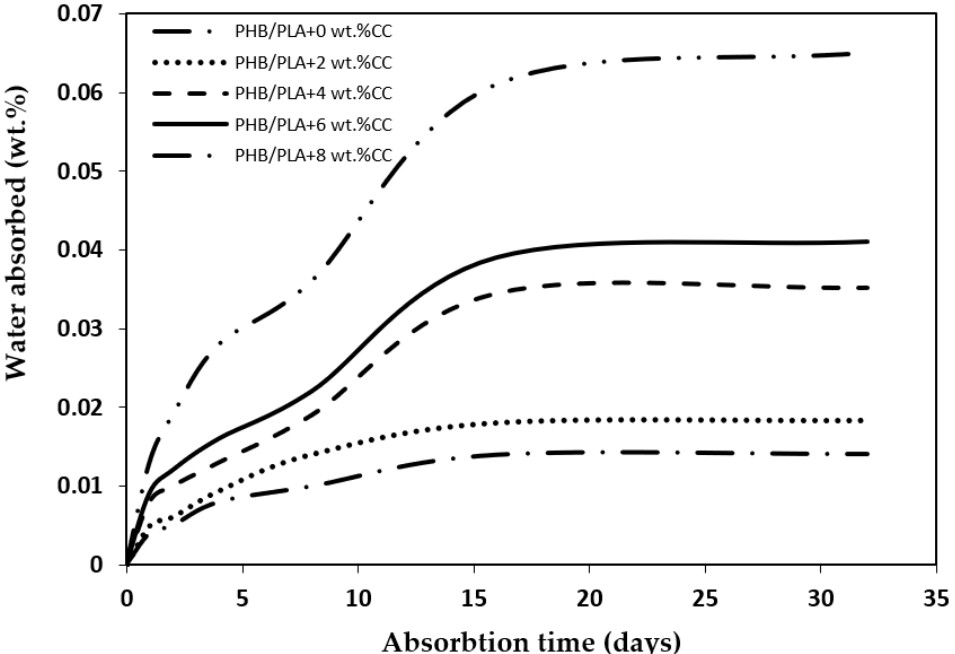

**Figure 14.** Effect of filler loading on water absorption of PHB/PLA corncob composites.

*3.10. Regression Analysis*

The mechanical properties evaluated within the specified filler loading range displayed a fourth-order polynomial response to filler loading increase. The following regression equations were established to define the different parameters under study, for instance, when filler loadings rose from 2 to 8 wt.%.

Filament tensile strength (preliminary test) ($FT_s$):

$$FT_s = -0.0406\,f^{\,4} + 0.6583\,f^{\,3} - 3.2125\,f^{\,2} + 3.2667\,f + 33.3 \quad (R^2 = 1) \tag{2}$$

Filament tensile modulus (preliminary test) ($FT_m$):

$$FT_m = -0.001\,f^{\,4} + 0.0118\,f^{\,3} - 0.0209\,f^{\,2} - 0.0321\,f + 1.4 \quad (R^2 = 1) \tag{3}$$

Tensile strength ($T_s$):

$$T_s = -0.00226\,f^{\,4} + 0.388\,f^{\,3} - 2.0059\,f^{\,2} + 2.4254\,f + 33.32 \quad (R^2 = 1) \tag{4}$$

Tensile modulus ($T_m$):

$$T_m = -0.0003\,f^4 + 0.0048\,f^3 - 0.03\,f^2 + 0.1383\,f + 2.99 \quad (R^2 = 1) \tag{5}$$

Flexural strength ($F_s$):

$$F_s = 0.0092\,f^4 - 0.0874\,f^3 - 0.0831\,f^2 + 0.5509\,f + 33.846 \quad (R^2 = 1) \tag{6}$$

Flexural modulus ($F_m$):

$$F_m = -0.0011\,f^4 - 0.015\,f^3 - 0.0569\,f^2 + 0.0975\,f + 2.61 \quad (R^2 = 1) \tag{7}$$

Charpy impact energy at 22 °C ($\mu_{22}$):

$$\mu_{22} = 0.0021\,f^4 - 0.0271\,f^3 + 0.0792\,f^2 - 0.2667\,f + 39.5 \quad (R^2 = 1) \tag{8}$$

Charpy impact energy at −38 °C ($\mu_{-38}$):

$$\mu_{-38} = 0.012\,f^4 - 0.2208\,f^3 + 1.2646\,f^2 - 2.4417\,f + 32.1 \quad (R^2 = 1) \tag{9}$$

where, $f$ is the filler loading in wt.% and $R^2$ is the statistical coefficient of determination (goodness of fit).

These regression equations are in-line with those generated for moisture content effects on dikanuts, a similar lignocellulosic biomaterial where all properties were found to have a polynomial (second order or third order) response to moisture content increase from 7.45% and 15.2% (dry basis) [73]. A study was conducted on the influence of filler loading on flowable resin and reported a linear regression [74], where the coefficients of determination generated were between 0.8 and 0.9. The results, and the coefficients of determination values obtained from this study, generally suggest that a polynomial regression would best correlate the effect of filler loading to the mechanical properties of composites.

### 3.11. Future Work

In an effort to improve the composite properties, an application of a surface treatment to the corncob powder could be conducted. For instance, maleic anhydride was shown to improve the bonding between PLA and lignin by surface polarity [75]. A silane coupling agent can also be added to the formulation, which has been reported to improve the adhesion between the hydrophilic natural based filler and hydrophobic PLA matrix [17]. In addition, the low toughness of PLA can be enhanced by the addition of a plasticizer to the formulation, and can improve the tensile strengths at higher filler loadings [15].

### 4. Conclusions and Recommendations

The engineering properties of 3D printed PHB/PLA composites samples with different corncob filler loadings of 0, 2, 4, 6, and 8 wt.% were assessed. The findings demonstrated that although the fillers enhanced the tensile and flexural moduli, adding particulate corncob fillers to the PHB/PLA bio-polymer blend often resulted in a reduction in tensile strength, flexural strength, and the Charpy impact strength. Values obtained for mechanical testing of samples with 0 wt.% corncob were all within the range of values reported in literature for injection molded pure PHB and pure PLA, concurring that 3D printing can be employed industrially as a viable substitute for the demanding injection molding process. The fractured tensile and flexural surfaces of unfilled PHB/PLA composite featured smooth surface characteristics suggestive of a brittle material, but the composites exhibited an increasing degree of roughness as filler loadings increased. Due to the high strain rate of the test, Charpy impact toughness fractured surfaces showed a predominantly flat surface for both filled and unfilled composites. Micrographs of both fractured tensile and flexural surfaces revealed a number of filler clusters at the fractured surface for higher filler loadings, due to agglomeration of filler particles. Composites with 8 wt.% corncob

absorbed the greatest amount of water, while unfilled PHB/PLA composites absorbed the least amount. The glass transition temperature increased as corncob filler loading increased from 0 to 8 wt. However, there was no significant difference between the $T_g$ of pure PHB/PLA samples and samples containing 2 wt.% corncob filler. Based on the findings of the Charpy impact tests, this composite material is not recommended for use in very cold weather applications that call for impact toughness. According to the study's findings, the PHB/PLA bio-polymer matrix used in 3D printing should only contain up to 6 wt.% (25 μm) of corncob for mechanical applications.

**Author Contributions:** Conceptualization, O.O. and D.C.; methodology, O.O. and D.C.; formal analysis, O.O.; investigation, O.O.; resources, D.C.; data curation, O.O and D.C.; writing—original draft preparation, O.O.; writing—review and editing, D.C.; supervision, D.C.; project administration and funding acquisition, D.C. All authors have read and agreed to the published version of the manuscript.

**Funding:** This research was funded by Natural Sciences and Engineering Research Council of Canada (NSERC) under the Discovery Grant (RGPIN-2020-06701). The APC was funded by MDPI.

**Data Availability Statement:** The data presented in this study are available on request from the corresponding author. The data are not publicly available because the raw and processed data required to reproduce these findings cannot be shared at this time, as the data also forms part of an ongoing study.

**Conflicts of Interest:** The authors declare no conflict of interest. The funders had no role in the design of the study; in the collection, analyses, or interpretation of data; in the writing of the manuscript; or in the decision to publish the results.

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
