# Peer review of "Development and Characterization of PHB-PLA/Corncob Composite for Fused Filament Fabrication"

_jcs, doi:10.3390/jcs6090249_

Round 1

Reviewer 1 Report

First of all, the manuscript was well-written with a well-organized structure. However, it presents a few of unclear points that may require the authors to further address.

1. The study is focused on the fused filament fabrication. The introduction doesn't include any background about the fused filament fabrication. The authors should introduce biopolymers and/or agricultural wastes that have been used in the fused filament fabrication, including the mechanical requirements, material selection, polymer properties, filler selection, etc., for filaments used for successful 3D printing processes. Compare them with the authors' hypotheses in the study and highlight the difference and the advantages of PHB/PLA/corncob powder over other materials.

2. The authors did not perform the compounding to product pellets prior to filament fabrication, instead, all the material components were added to filament extruder to product filament directly. How did the author ensure the mixing homogeneity? Was the filler content same in any fragment of filament? Maybe ash content test results from different filament samples will tell. In addition, the nonuniform filament would also cause the mechanical defects, which will lead to adverse mechanical properties.

3. Some typical properties of polymer composites after extrusion should be given, such as melt flow and moisture content.

Reviewer 2 Report

- The approach is interesting and the topic is appropriate for the journal.

-        The work  has a very clear structure and all the sections are well written in a way that is easy to read and understand.

-         However, little modifications and improvements are needed to enhance the quality of the paper.

-        The paper is focused on the Development and Characterization of PHB-PLA/Corncob Composite for Fused Filament Fabrication, reporting interesting results. In the “Introduction” section, the authors start to discuss about  “environmental difficulties as a result of the widespread usage of non-biodegradable materials, particularly plastics “ as well as on the potential use of bodegradable polymers and advanced processing techniques (e.g., additive manufacturing/3D printing ). Even though the authors already report some strategies in literature related to topic and they especially focus on PHB-PLA/Corncob Composite for Fused Filament Fabrication, concerning the Introduction section I also suggest to BRIEFLY report further concepts and results in the development of 3D printed parts using virgin and recycled biodegradable polymer matrix as well as methodological analyses to provide further insights and to individuate a viable option  (e.g., Procedia CIRP 79 (2019), 143-146 …). Then, the authors should continue to stress their study related to PHB-PLA/Corncob Composite for Fused Filament Fabrication. All of this should improve the quality of the paper, reporting important features as well as further methodologies  in the development of 3D printed devices according to the environmental impact benefiting from recycling concepts and biodegradable polymers, thus helping the different kinds of readers to better understand the value of their work.

-        The Introduction and/or discussion section as well as the list of references should be improved according to the above reported comments.

-        The quality of some figures should be improved.

-        The title is adequate and appropriate for the content of the article.

-        The abstract contains information of the article.

-        Figures and captions are essential and clearly reported.

Reviewer 3 Report

The author Ohaeri et al. developed a PHB-PLA/Corncob full biobased composite material for extrusion type 3D printing applications. In terms of the scope, the work fits within the Journal of Composite Science. It is also important to explore the full potential of biocomposite to make it further down the road of applications. However, there are some comments that must be addressed before this work can be considered for publication.

  1. In most PLA and PHB research, people often focus on the toughening of the materials. However, in this case, it appears brittleness is the obvious effect. Is there any particular reason why this is the case?
  2. Have the authors considered any surface treatment method or compatibilizer for the composite (Future work?)? Plant-based cellulose as an additive often makes the materials harder (higher modulus) but more brittle (lower toughness). Many treatments method can address this issue (maleic anhydride…etc).
  3. Any particular reason why the 55/45 ration of PGB/PLA was selected?
  4. Line 151, please indicate the company and origin of the SolidWorks software.
  5. ASTM D638 has 5 different types of dogbone, which one was used in this case?

Reviewer 4 Report

The manuscript deals with the development of PHB-PLA blends filled with different ratios of corncob powder to be directly used as filaments for 3D printing process. In addition, several mechanical tests, thermal analysis, water absorption and morphological analysis were employed to characterize the developed biocomposites.

The work brings some new findings to the field, however and at this stage there are some points that should be addressed, namely:

In the manuscript it is missing an image showing the developed 3D printed materials. The author can also provide the information in the graphical abstract.

Page4, line 128: In the methodology to create the composite filament, it is not clear if the authors make a two-step extrusion process, being in the first step to create the blend and in the second extrusion step to create the composite filament. Moreover, it is missing the processing temperature used and more detail on the extruder machine used, such as mono-screw extruder.

Page 8, table 3: Micrographs of a section of the filament at higher magnification should be provided in order to provide the discussion of the author, namely when occurs voids in the filament or to better discuss the filler matrix interaction.

Page 9, line 272: Regarding the tensile properties, the tensile strain (%) properties of the specimens can be provide and discussed by the author.

Page 15, line 521: In the DSC analysis the author states that the addition of corncob filler enhances the Tg of the pure PHB/PLA, however and based on the obtained results the author needs to provide the reason for that behavior in the presence of corncob filler.
